# Soluble Expression of Fc-Fused T Cell Receptors Allows Yielding Novel Bispecific T Cell Engagers

**DOI:** 10.3390/biomedicines9070790

**Published:** 2021-07-08

**Authors:** Wen-Bin Zhao, Ying Shen, Wen-Hui Liu, Yi-Ming Li, Shi-Jie Jin, Ying-Chun Xu, Li-Qiang Pan, Zhan Zhou, Shu-Qing Chen

**Affiliations:** Institute of Drug Metabolism and Pharmaceutical Analysis & Zhejiang Provincial Key Laboratory of Anti-Cancer Drug Research, College of Pharmaceutical Sciences, Zhejiang University, Hangzhou 310058, China; pharmacy_zwb@zju.edu.cn (W.-B.Z.); shinesysy@foxmail.com (Y.S.); liuwenhui@zju.edu.cn (W.-H.L.); yiming_li@zju.edu.cn (Y.-M.L.); jinsj08@163.com (S.-J.J.); ycxu66@163.com (Y.-C.X.); panliqiang@zju.edu.cn (L.-Q.P.)

**Keywords:** T cell receptor, soluble expression, bispecific T cell engagers, NY-ESO-1/LAGE-1, staphylococcal enterotoxin C2

## Abstract

The specific recognition of T cell receptors (TCR) and peptides presented by human leukocyte antigens (pHLAs) is the core step for T cell triggering to execute anti-tumor activity. However, TCR assembly and soluble expression are challenging, which precludes the broad use of TCR in tumor therapy. Herein, we used heterodimeric Fc to assist in the correct assembly of TCRs to achieve the stable and soluble expression of several TCRs in mammalian cells, and the soluble TCRs enable us to yield novel bispecific T cell engagers (TCR/aCD3) through pairing them with an anti-CD3 antibody. The NY-ESO-1/LAGE-1 targeted TCR/aCD3 (NY-TCR/aCD3) that we generated can redirect naïve T cells to specific lysis antigen-positive tumor cells, but the potency of the NY-TCR/aCD3 was disappointing. Furthermore, we found that the activation of T cells by NY-TCR/aCD3 was mild and unabiding, and the activity of NY-TCR/aCD3 could be significantly improved when we replaced naïve T cells with pre-activated T cells. Therefore, we employed the robust T cell activation ability of staphylococcal enterotoxin C2 (SEC2) to optimize the activity of NY-TCR/aCD3. Moreover, we found that the secretions of SEC2-activated T cells can promote HLA-I expression and thus increase target levels, which may further contribute to improving the activity of NY-TCR/aCD3. Our study described novel strategies for soluble TCR expression, and the optimization of the generation and potency of TCR/aCD3 provided a representative for us to fully exploit TCRs for the precision targeting of cancers.

## 1. Introduction

In recent years, T cell-based immunotherapies, such as immune checkpoint inhibitors (ICIs) [1], adoptive cell therapy (ACT) [2,3], and cancer vaccines [4,5], have emerged as promising strategies for the treatment of cancer and have led to clinical remission in some patients. However, the specificity and efficiency of these immunotherapies have been unable to balance effectively. ICIs have exhibited dramatic anti-tumor efficacy in patients with certain cancer types, while the non-discriminatory sensitization of the T cells caused by ICIs comes with immune-related adverse effects [6]; the Chimeric antigen receptor (CAR) T cell can be highly effective against leukemias and lymphomas, while toxicities (e.g., cytokine release syndrome, neurotoxicity, B cell aplasia) have often been observed in patients who have received CAR T therapies within days of the first infusion [7]. To a large extent, the adverse effects of ICIs and CAR T therapies were caused by the deficient target specificity [3,6]. Peptides, which are derived from tumor-specific proteins (e.g., somatic mutations, carcino-embryonic antigens, tumorigenic virus antigens), are highly restricted to tumor cells [8]. Specific peptide-based cancer vaccines have been proved to be feasible and safe, although the mild effect of these cancer vaccines makes it difficult for these vaccines to meet clinical needs [9,10]. Therefore, novel immunotherapeutic strategies that simultaneously possess exquisite tumor specificity and high activity urgently need to be explored.

Peptides presented by human leukocyte antigen (pHLA) can be recognized by specific T cell receptors (TCR) that then trigger T cell immune response [11]. The adoptive transfer of ex vivo expanded autologous tumor-infiltrating lymphocytes (TILs) has demonstrated dramatic clinical efficacy [12,13], which indicates that the sufficient quantity and the effective activation of specific T cells are needed for tumor regression. Although there are huge numbers of T cells in tumor patients, most of them are passengers to tumors. In addition, the rapid ex vivo expansion of functional T cells remains a challenge. Bispecific T cell engagers (BiTEs) can indiscriminately endow T cells with tumor-targeting ability by binding CD3ε that fully mobilize T cells to kill tumors [14]. To take the advantage of TCRs and BiTEs, Liddy N et al. [15] and Middleton MR et al. [16] developed ImmTACs, which comprised tumor-associated monoclonal TCR fused to a CD3ε-specific antibody (TCR-based BiTE, TCR/aCD3), and the ImmTACs showed exquisite potency and in vivo efficacy against tumors. Nevertheless, that generation of ImmTAC was based on protein refolding after being expressed as inclusion bodies in *Escherichia coli*. The insufficient refolding efficiency, endotoxins residues, and complex manipulation of purification posed a challenge to the preparation of the accredited TCR/aCD3 [17]. Thus, the exploration of safe, efficient, and handy technology has a significant role in promoting the clinical application of TCR/aCD3.

Mammalian cells are the most attractive tool for the production of protein drugs [18], and TCRs can be expressed in mammalian cells in their soluble form [19,20]. Specifically, Froning K, et al. [19] produced TCR/aCD3 in mammalian cells that provided a novel strategy for TCR/aCD3 generation, but the biological activity of this mammalian cell-produced TCR/aCD3 and the universality of this technology have not been well studied. Inspired by this research, we achieved the soluble expression of the two components of TCR/aCD3 (TCRs and CD3ε-specific antibody) in mammalian cells, and click chemical reaction was then used to re-assemble the two components to generate TCR/aCD3. The CD3ε-specific antibody (aCD3) is an invariable component of TCR/aCD3, so it is flexible enough to generate diverse TCR/aCD3 by altering TCR. In addition, we used NY/A02 (NY-ESO-1/LAGE-1_157–165_ epitope in the context of HLA-A*0201) targeting TCR (named as NY-TCR) [21] as a representative to evaluate the biological activity of the novel TCR/aCD3 here. The NY-TCR/aCD3 showed the ability to activate and redirect T cells to specifically kill tumor cells. However, the potency of the NY-TCR/aCD3 was not satisfied as expected, even though the affinity of NY-TCR had been tremendously improved. We speculated that inadequate T cell activation may be the main factor leading to the insufficient activity of NY-TCR/aCD3. Interestingly, we found that the T cell activation property of staphylococcal enterotoxin C2 (SEC2) [22] can be employed to improve the cytotoxic potency of NY-TCR/aCD3 thus uniting two such biologic components together might markedly enhance naïve T cell response against tumors.

## 2. Materials and Methods

### 2.1. Cell Culture

Human embryonic kidney 293 cells (HEK293 cells) were kindly provided by the Comprehensive AIDS Research Center (Tsinghua University) and were maintained in SMM 293-TI medium (Sino Biological Inc., Beijing, China) supplemented with 0.5% fetal bovine serum (FBS; Gibco, Grand Island, NY, USA) and 1% penicillin-streptomycin and were rotary incubated at 120 rpm at 37 °C with 5% CO_2_ and 95% humidity. A375 cells (RRID: CVCL_0132), K562 cells (RRID: CVCL_0004), Jurkat cells (RRID: CVCL_0367), and T2 cells (RRID: CVCL_2211) were purchased from American Type Culture Collection (ATCC), and K562-A2 cells (HLA-A*0201 transfected K562 cells) were constructed and kept in the laboratory. All of the tumor cells used here were maintained in RPMI-1640 medium (Gibco, Grand Island, NY, USA) supplemented with 10% FBS and 1% penicillin-streptomycin (complete RPMI-1640 medium) at 37 °C with 5% CO_2_ and 95% humidity. T2 cells were maintained in IMDM medium (Gibco, Grand Island, NY, USA) supplemented with 20% FBS and 1% penicillin-streptomycin at 37 °C with 5% CO_2_ and 95% humidity.

### 2.2. Soluble Expression of TCRs and aCD3 in HEK293 Cells

The sequences of the TCRs were obtained from previous studies [21,23]. Fc fused TCR genes (tagged with sortase A recognition sequence) were synthesized using the Sangon Biotech (Shanghai) Co., Ltd. and then inserted into the pMH3 plasmid (AmProtein China Cooperation Ltd., Hangzhou, China) to construct the expression plasmid. HEK293 cells in the logarithmic growth phase were transfected with the expression plasmids and cultured for 4 days according to the method mentioned in the cell culture. The supernatant of the HEK293 cells was then collected and the TCRs were purified using protein A column (HiTrap^TM^ Protein A HP, GE Healthcare Life Sciences, Little Chalfont, Buckinghamshire, USA). The purified TCRs were analyzed using SDS-PAGE in denatured condition after being de-glycosylated using PNGase F (Sigma-Aldrich, St. Louis, MO, USA).

The sequences of the variable domain of the aCD3 were obtained from a previous study [24]. The aCD3 genes (Fab, tagged with *(G*_4_*S)*_3_-LPETG-6×His at the C-terminus of the heavy chain) were synthesized using the Sangon Biotech (Shanghai) Co., Ltd. (Shanghai, China) and were then inserted into the pMH3 plasmid to construct the expression plasmid. The aCD3 was expressed using HEK293 cells according to the method mentioned for TCR expression and purified using Ni column (HisTrap^TM^ HP, GE Healthcare Life Sciences, Little Chalfont, Buckinghamshire, UK).

### 2.3. Binding of NY-TCRmut to T2 Cells Loaded with Peptides

T2 cells can be loaded with exogenous peptides and β2-microglobulin (β2m) to form specific pHLA on the cell surface. For peptide pulsing, T2 cells in the logarithmic growth phase were cultured at 5 × 10^5^ cells/mL in serum-free IMDM (Gibco, Grand Island, NY, USA) containing 10 μg/mL β2m (Sigma-Aldrich, St. Louis, MO, USA) and either 30 μg/mL irrelevant peptide RMFPNAYL or relevant peptide SLLMWITQC (synthesized by the Sangon Biotech (Shanghai) Co., Ltd., Shanghai, China) for 6–8 h at 37 °C with 5% CO_2_ and 95% humidity. For the cell-binding assay, peptide-pulsed T2 cells were collected and resuspended in 100 μL ice-cold staining buffer (1% BSA in PBS) and then mixed with 1 μg NY-TCRmut (affinity-enhanced NY-TCR). After incubation on ice for 30 min, cells were washed twice with staining buffer, and FITC labelled goat anti-human IgG (H + L) (Beyotime Biotechnology, Shanghai, China) were then added and incubated on ice for 30 min. Finally, cells were examined by flow cytometry (NovoCyte^TM^, ACEA Biosciences, San Diego, CA, USA) after being washed with staining buffer.

### 2.4. Flow Cytometry Analysis for Cell-Binding Assays

To assess the binding affinity of NY-TCRs and NY-TCR/aCD3 on cells, approximately 5 × 10^5^ cells were collected and resuspended in ice-cold PBS and mixed with different concentrations of either NY-TCRs or NY-TCR/aCD3 on ice for about 30 min. Cells were then washed twice with ice-cold PBS, immunostained with PE labelled goat anti-human IgG Fc (ThermoFisher, Waltham, MA, USA) on ice for about 30 min and were then analyzed using flow cytometry after being washed twice with ice-cold PBS.

### 2.5. Generation of NY-TCR/aCD3

Sortase A was generated as it had been in a previous report [25]. Sortase A reactions were performed in 50 mM Tris-HCl (pH = 7.4), 150 mM NaCl, and 5 mM CaCl_2_ and consisted of either NY-TCR (2 μM) or aCD3 (6 μM), Sortase A (50 μM), and GGG-PEG_3_-N_3_ (200 μM synthesized by the Levena Biopharma Co., Ltd. (Suzhou, China)) or GGG-PEG_4_-DBCO (200 μM synthesized by the Levena Biopharma Co., Ltd. (Suzhou, China)). The reactions were incubated at 37 °C for 12 h, and NY-TCR-N_3_ was then purified using protein A column, and aCD3-DBCO was purified using Ni column. Click reaction was performed in 50 mM Tris-HCl (pH = 7.4), 150 mM NaCl and consisted of NY-TCR-N_3_ (50 μM) and aCD3-DBCO (50 μM). The reaction was rotated overnight at 4 °C, and the NY-TCR/aCD3 was then purified by means of molecular exclusion chromatography (Superdex^TM^ 200 Increase 10/300 GL, GE Healthcare Life Sciences, Little Chalfont, Buckinghamshire, USA). The products were analyzed using SDS-PAGE in non-denatured condition or SDS-PAGE in denatured condition after being de-glycosylated using PNGase F.

### 2.6. Enzyme-Linked Immunosorbent Assay (ELISA) for Affinity and Specificity Analysis of NY-TCR/aCD3

Biotin-labelled pHLAs were generated through refolding as per the previous report [26]. A 96-well EIA/RIA plate (Corning Incorporated, Corning, NY, USA) was coated with streptavidin (2 μg/mL, 0.1 mL) in coating buffer (0.5 M carbonate/bicarbonate buffer, pH = 9.6) at 4 °C overnight. After incubation, the plate was washed with PBST (PBS containing 0.05% Tween 20) 3 times and was then blocked with blocking buffer (5% skim powder in PBST) at 37 °C for 1 h. The wells were then incubated with different biotin-labelled pHLAs at 37 °C for 1 h. The plate was then washed 4 times with PBST before a 1 h incubation with different concentrations of NY-TCR/aCD3 at 37 °C followed by a 0.5 h incubation with HRP-conjugated goat anti-human IgG (H + L) (Beyotime Biotechnology, Shanghai, China) at 37 °C after 4 washes with PBST. Finally, the samples were detected using tetra-methyl-benzidine (TMB, Sangon Biotech (Shanghai) Co., Ltd. (Shanghai, China)) after 6 washes with PBST, and the reaction was stopped using 2 M H_2_SO_4_ followed by the measuring the absorbance at 450 nm. The results were presented as the mean ± S.D (n = 3)

### 2.7. Preparation of Effector Cells

Human peripheral blood mononuclear cells (PBMC) were purchased from SAILY Biotechnology (Shanghai) Co., Ltd. (Shanghai, China) To obtain pre-activated T cells (ATC), 5 × 10^6^ PBMC were resuspended in 10 mL complete RPMI-1640 medium containing 30 U/mL human IL-2 (R&D Systems, Minneapolis, MN, USA) and 1 × 10^7^ human T-activator CD3/CD28 beads (Gibco, Grand Island, NY, USA) at 37 °C with 5% CO_2_ and 95% humidity for 5–7 days.

### 2.8. Evaluation of T Cell Activation Stimulated by NY-TCR/aCD3

3 × 10^4^ A375 cells and 1.2 × 10^5^ PBMC were co-cultured in 0.3 mL complete RPMI-1640 medium containing different concentrations of NY-TCR/aCD3 at 37 °C with 5% CO_2_ and 95% humidity for 48 h or 72 h. After incubation, the supernatant was collected to determine the concentration of interferon-γ (IFN-γ) using the IFN-γ human ELISA kit (ThermoFisher, Waltham, MA, USA). Cells were resuspended in ice-cold PBS and immunostained with FITC labeled mouse anti-human CD4 or CD8 antibody (ThermoFisher, Waltham, MA, USA), pacific blue labeled mouse anti-human CD69 antibody (ThermoFisher, Waltham, MA, USA), and APC labeled mouse anti-human CD25 antibody (ThermoFisher, Waltham, MA, USA) on ice for about 30 min and then analyzed using flow cytometry after being washed twice with ice-cold PBS. The results were presented as the mean ± S.D (n = 3).

### 2.9. Evaluation of the Anti-Tumor Activity of NY-TCR/aCD3 In Vitro

For the tumor cell apoptosis assay, 3 × 10^4^ eGFP^+^ tumor cells (A375-eGFP, A375-NY, K562-NY, or K562-Ctrl, Appendix A) were co-cultured with 1.2 × 10^5^ PBMC or 7.5 × 10^4^ ATC in 0.3 mL complete RPMI-1640 medium containing different concentrations of NY-TCR/aCD3 at 37 °C with 5% CO_2_ and 95% humidity for 24 h (ATC) or 48 h (PBMC). After incubation, cells were collected and labelled with fluorescence (15 min at room temperature) using Annexin V, 633 Apoptosis Detection Kit (Dojindo, Kumamoto, Japan) and then analyzed using flow cytometry (eGFP fluorescence was used to distinguish tumor cells from effector cells) to determine the tumor cell apoptosis ratio. The corrected apoptosis ratio = the apoptosis ratio of the test group-the apoptosis ratio of the control group (0 ng/mL NY-TCR/aCD3). The results were presented as the mean ± S.D (n = 3).

For the lactate dehydrogenase (LDH) assay, 1.5 × 10^4^ tumor cells were co-cultured with 6 × 10^4^ PBMC or 3.75 × 10^4^ ATC in 160 μL phenol red-free RPMI-1640 medium (Gibco, Grand Island, NY, USA) containing 2% FBS, 1% penicillin-streptomycin, and different concentrations of NY-TCR/aCD3 at 37 °C with 5% CO_2_ and 95% humidity for 24 h (ATC) or 48 h (PBMC). After incubation, the relative concentration of LDH in the supernatant (OD_490_ value) was determined using the LDH Cytotoxicity Assay Kit (Beyotime Biotechnology, Shanghai, China). The corrected LDH released level = OD_490_ value of the test group/OD_490_ value of the maximum lysis group×100%. The results were presented as the mean ± S.D (n = 3).

For the competition inhibition assay, 1.5 × 10^4^ A375-NY cells and 3.75 × 10^4^ ATC were co-cultured in 160 μL phenol red-free RPMI-1640 medium as above, which contained 100 ng/mL NY-TCRmut/aCD3 and different concentrations of NY/A02-Biotin at 37 °C with 5% CO_2_ and 95% humidity for 24 h. After incubation, the relative concentration of LDH in the supernatant was determined using the LDH Cytotoxicity Assay Kit. Competition inhibition = OD_490_ value of the test group/OD_490_ value of the control group (0 ng/mL NY/A02) × 100%. The results were presented as the mean ± S.D (n = 3).

### 2.10. Evaluation of the Anti-Tumor Activity of NY-TCR/aCD3 In Vivo

Eight-week-old female beige-SCID mice (Charles River Laboratories Co., Ltd., Beijing, China) were subcutaneously co-engrafted with A375 cells (2 × 10^6^) and ATC (4 × 10^6^) in the right armpit. The mice were randomly divided into four groups and were then treated intravenously with NY-TCRmut/aCD3 (1 mg/kg or 5 mg/kg), aCD3 (1.5 mg/kg, the molality was similar to 5 mg/kg NY-TCRmut/aCD3), and saline 1 h after engraftment as well as 4 days later for the second dose (q4d × 2). After the first administration, tumor length and width were measured every 4–5 days, and the tumor volume was calculated according to the formula (tumor volume = length × width^2^/2). The results were presented as the mean ± S.D (n = 5). Student’s *t*-test was used to determine statistical significance (* *p* < 0.05, ** *p* < 0.01, *** *p* < 0.001, n.s *p* > 0.05).

### 2.11. Analysis Expression Level of NY/A02 after Stimulated by SEC2

SEC2-His was generated as it was the previous report [22]. 1 × 10^7^ PBMC were cultured with or without 1 μg/mL SEC2-His in 10 mL complete RPMI-1640 medium at 37 °C with 5% CO_2_ and 95% humidity for 48 h. After incubation, the supernatant was collected. For the HLA-A*0201 expression assay, A375 cells were cultured in one of the following mixtures: the 1 mL supernatant from PBMC and 1 mL complete RPMI-1640 medium (PBMC group); 1 mL supernatant from the SEC2-activated T cells and 1 mL complete RPMI-1640 medium (PBMC/SEC2 group); 2 mL complete RPMI-1640 medium containing 0.5 μg/mL SEC2-His (SEC2 group); or 2 mL complete RPMI-1640 medium (control), each cultured at 37 °C with 5% CO_2_ and 95% humidity for 48 h. After incubation, A375 cells were resuspended in ice-cold PBS and immunostained with FITC labeled BB7.2 (HLA-A*02 monoclonal antibody, ThermoFisher, USA) on ice for about 30 min and followed by flow cytometry analysis after being washed twice with ice-cold PBS. For the NY/A02 expression assay, A375-NY cells were cultured in the 1 mL of the supernatant from the PBMC and 1 mL of complete RPMI-1640 medium (PBMC group); 1 mL of supernatant from the SEC2-activated T cells and 1 mL of complete RPMI-1640 medium (PBMC/SEC2 group); 2 mL of complete RPMI-1640 medium containing 0.5 μg/mL SEC2-His (SEC2 group); or 2 mL complete RPMI-1640 medium (control), each cultured at 37 °C with 5% CO_2_ and 95% humidity for 48 h. After incubation, the A375-NY cells were resuspended in ice-cold PBS with 10 μg/mL NY-TCRmut on ice for about 30 min and immunostained with PE labelled goat anti-human IgG Fc on ice for about 30 min after being washed twice with ice-cold PBS, followed by flow cytometry analysis and being washed again with ice-cold PBS.

### 2.12. Evaluation of the Combined Anti-Tumor Activity of NY-TCRmut/aCD3 and SEC2

3 × 10^4^ eGFP^+^ tumor cells were co-cultured with 1.2 × 10^5^ PBMC in 0.3 mL complete RPMI-1640 medium containing different concentrations of NY-TCRmut/aCD3 and 2 ng/mL or 0 ng/mL SEC2-His at 37 °C with 5% CO_2_ and 95% humidity for 48 h. Tumor cell apoptosis was determined as described above. Student’s *t*-test was used to determine statistical significance.

## 3. Results

### 3.1. Soluble Expressions of TCRs and aCD3 in Mammalian Cells

Two Fc-fused formats were designed to produce soluble TCRs in HEK293 cells (Figure 1A, tagged with *G_4_S*-LPETGG at C-terminus), and NY-TCRmut was able to be successfully expressed in both formats. For both formats of NY-TCRmut, three bands appeared above the theoretical position on the reducing SDS-PAGE gel after purification, and the bands were restored to the proper size after deglycosylation, which likely reflected the partial glycan occupancy of the N-glycosylation sites (Figure 1B). Nevertheless, the purity of the NY-TCRmut was high, and this was also verified by RP-HPLC (Appendix A). We then assayed the antigen-binding ability of the NY-TCRmut on T2 cells and found that both of them could specifically recognize T2 cells incubated with the relevant peptide SLLMWITQC and they were able to ignore T2 cells incubated with the irrelevant peptide RMFPNAPYL, indicating the specific binding of the NY-TCRmut to NY/A02 (Figure 1C). Moreover, both NY-TCRmut exhibited a similar binding ability to A375-NY cells (NY/A02 positive, Figure 1D). Although the properties of the two formats of NY-TCRmut are similar, the expression level of format 1 (~20 mg/L) is significantly higher than that of format 2 (~6 mg/L), which indicated that format 1 may be the preferred option. The soluble aCD3 (Fab, tagged with (*G*_4_*S*)_3_-LPETG-6×His at the C-terminus of the heavy chain) was expressed in HEK293 cells as well.

To investigate if format 1 is feasible for the soluble expression of diverse TCRs, we attempted to express soluble Tax-TCR (binding to HTLV Tax_11–19_ epitope in the context of HLA-A*0201), p53^R175H^-TCR (binding to residues 168-176 of p53^R175H^ in the context of HLA-A*0201), and NY-TCRwt in HEK293 cells. All three TCRs were successfully expressed and analyzed by reducing SDS-PAGE gel after deglycosylation. The result showed that the three TCRs as well as NY-TCRmut were glycosylated (Figure 1E).

### 3.2. Generation of NY-TCR/aCD3

The NY-TCRmut and NY-TCRwt of format 1 were selected to generate TCR/aCD3 (Figure 2A). Sortase A-mediated transpeptidation was used to specifically conjugate GGG-PEG_3_-N_3_, GGG-PEG_4_-DBCO to NY-TCR (NY-TCR-N_3_), and aCD3 (aCD3-DBCO, Appendix A), respectively. After removing the dissociative small molecules and sortase A, we assayed the conjugated products using RP-HPLC. The conjugation efficiency between GGG-PEG_4_-DBCO and aCD3 was high; however, the conjugation efficiency between GGG-PEG_3_-N_3_ and NY-TCR was difficult to confirm (Appendix A).

NY-TCR/aCD3 was then generated through click-reaction (N_3_ and DBCO) after purification using molecular exclusion chromatography (Figure 2B). Through SDS-PAGE analysis in the non-denatured condition, we preliminarily determined that Peak 2 was the expected product with two bands (one NY-TCR conjugated with one or two aCD3, Figure 2C). We further found that the product of Peak 2 showed three bands (α or β chain of NY-TCR, α or β chain of NY-TCR conjugated with the heavy chain of aCD3, and the light chain of aCD3) of theoretical size after being deglycosylated using PNGase F during SDS-PAGE analysis with the denatured condition (Figure 2D), and it can simultaneously recognize Jurkat cells (CD3ε^+^) and soluble NY/A02 (Appendix A), which indicated that the product of Peak 2 was NY-TCR/aCD3.

Finally, we compared the binding ability of NY-TCRmut/aCD3 and NY-TCRwt/aCD3 to the targets and found that they had a similar binding ability to Jurkat cells (Figure 2E). Additionally, NY-TCRwt/aCD3 had a weak affinity to NY/A02, while the affinity between NY/A02 and NY-TCRmut/aCD3 was significantly higher (Figure 2F). Moreover, NY-TCRmut/aCD3 did not show any affinity to irrelevant pHLA under otherwise identical conditions, indicating the specific binding of the NY-TCRmut/aCD3 to NY/A02 (Figure 2F).

### 3.3. NY-TCR/aCD3 Activated and Redirected Naïve T Cells to Specifically Kill NY/A02^+^ Tumor Cells

We used several methods to measure the T cell activation ability of NY-TCR/aCD3 in vitro. First, cell surfaces CD69 (early marker) and CD25 (intermediate or late marker) were chosen as the indicators of T cell activation. After co-incubation with A375 cells (NY-ESO-1^+^ and HLA-A*0201^+^, Appendix A) and different concentrations of NY-TCR/aCD3 for 72 h, the CD69 or CD25 surface expression of T cell was determined. We found that CD69^+^ or CD25^+^ T cells significantly increased with different concentrations of NY-TCR/aCD3, but they decreased with high concentrations of NY-TCR/aCD3 (≥2 μg/mL, Figure 3C). To test whether the CD69^+^ or CD25^+^ T cells were increased in a time-dependent manner, we performed the same T cell assay after a treatment with NY-TCRmut/aCD3 for 48 h and found that the CD69^+^ T cell ratio was similar to that of 72 h, and the CD25^+^ T cell ratio slightly increased (Appendix A). We then measured the T cell activation through IFN-γ assays. After co-incubation with A375 cells and different concentrations of NY-TCR/aCD3 for either 48 h or 72 h, the concentrations of IFN-γ in the supernatant were determined. Similar to the results of the CD25 expression assays, IFN-γ secretion significantly increased in a time-dependent manner, but it decreased at 10 μg/mL (Figure 3D). NY-TCRmut/aCD3 induced more IFN-γ secretion than NY-TCRwt/aCD3 in 72 h, which indicated that NY/A02 and NY-TCR interaction was an essential step in the T cell activation. Additionally, T cell activation was also supported by the proliferation of PBMC (Appendix A). In all of the cases above, the T cells were activated in the presence of antigen-positive tumor cells and NY-TCR/aCD3; however, the activation of T cells was not satisfied.

To further investigate the activity of NY-TCR/aCD3, we examined the ability of NY-TCR/aCD3 to mediate the specific killing of NY/A02^+^ tumor cells with T cells. The NY-TCRmut/aCD3 redirected T cells to kill A375 or A375-eGFP cells (A375 cells transfected with eGFP, Appendix A) in vitro, and the apoptosis ratios were below 20%, and the half-maximal effective concentration (EC_50_) value was 392 ng/mL (Figure 3A). Notably, NY-TCRwt/aCD3 was less active than NY-TCRmut/aCD3 (Figure 3A,B), which indicated that the affinity of NY-TCR is essential for the activity of NY-TCR/aCD3. Two cell lines derived from K562 cells (NY-ESO-1^-^ and HLA-A*0201^-^) were then used to investigate the specificity of NY-TCRmut/aCD3. As Figure 3B shows, NY-TCRmut/aCD3 can redirect T cells to kill K562-NY cells (NY/A02^+^, Appendix A), and negative control cells (K562-Ctrl, NY/A02^~^, Appendix A) were not killed even at the highest concentration used. These data confirm that NY-TCR/aCD3 can induce NY/A02^+^ cell killing; however, the lytic activity of NY-TCR/aCD3 induced T cells was mild.

### 3.4. NY-TCR/aCD3 Redirected ATC to Specifically Kill NY/A02^+^ Tumor Cells

To determine whether the insufficient activation of T cells was a limitation of the NY-TCR/aCD3 activity, ATC was obtained to replace the naïve T cells. We found that NY-TCRmut/aCD3 redirected ATC to kill A375 and A375-eGFP cells effectively, with the highest lytic level being 40%, and the EC_50_ value was 2 times lower (Figure 4A). Moreover, the highest lytic level of K562-NY cells induced by ATC and NY-TCRmut/aCD3 was more than 60%. Similar to the result above, NY-TCRwt/aCD3 showed about 10 times less effectiveness than NY-TCRmut/aCD3, and K562-Ctrl cells were not killed, even at the highest concentration of NY-TCR/aCD3 used (Figure 4A,C and Appendix A). Additionally, the cell killing of A375-NY can be almost completely inhibited by soluble NY/A02, further indicating the high specificity of NY-TCRmut/aCD3 (Figure 4B and Appendix A). These findings illustrate that inadequate T cell activation may be the stumbling block to NY-TCRmut/aCD3 activity.

To explore the activity of NY-TCRmut/aCD3 in vivo, we used xenograft models in which we engrafted beige-SCID mice subcutaneously with A375 cells and ATC. We treated the mice intravenously with NY-TCRmut/aCD3 (1 mg/kg or 5 mg/kg), aCD3 (1.5 mg/kg, the molality was similar to 5 mg/kg NY-TCR/anti-CD3, control group), or saline (blank group). The NY-TCRmut/aCD3 significantly inhibited tumor growth over the study period of 34 days compared to the control group, with the high dose resulting in significant reductions of tumor size compared to the low dose (Figure 4D).

### 3.5. SEC2 Enhanced the Anti-Tumor Activity of NY-TCRmut/aCD3

According to our findings above, we employed the robust T cell activation ability of SEC2 to optimize the activity of NY-TCRmut/aCD3. First, we confirmed that T cells were activated in the presence of SEC2, even at low concentrations (Appendix A). NY-TCRmut/aCD3 was able to redirect the lytic activity of T cells more effectively at different concentrations within 2 ng/mL SEC2 (Figure 5C and Appendix A). Interestingly, we found that the HLA-A*02 and NY/A02 expression levels were notably up-regulated after stimulation from the secretory substances of the SEC2-activated T cells in vitro (Figure 5A,B and Appendix A) and that SEC2 can promote the expression of pHLA in vivo (Appendix A), which may be another contribution of SEC2 to NY-TCRmut/aCD3 activity optimization.

## 4. Discussion

A benefit of the stringent selection processes during thymic development, natural TCR-pHLA interaction holds the exquisite specificity (self-tolerance) and sensitivity that enables T cell triggering to eliminate xenobiotics [27]. However, TCR assembly and soluble expression are challenging. Herein, we used heterodimeric Fc to assist in the correct assembly of TCRs to achieve the soluble expression of TCRs in mammalian cells. Compared to bacteria-produced TCRs, the Fc-fused TCRs described here are safe, convenient to operate, and expressed at a high level, which dramatically increased in their ease of production. These mammalian cell-produced TCRs also displayed the glycosylation that is essential for natural TCRs to execute their function [28]. In addition, glycosylation has been shown to stabilize and solubilize many different proteins including antibody Fc and Cα of TCRs, which may be one of the important reasons for the high expression of Fc-fused TCRs in mammalian cells [29,30].

Generally, more than 90% of cellular proteins have peptides that can be potentially presented through HLA to form pHLA, which can be targets of TCRs. In comparison, less than 10% of these proteins are validated to be directly present on the cell surface and could be recognized by traditional antibodies [31]. Therefore, TCRs have a vast target library, larger than that of traditional antibodies. Interestingly, tumor cell surface pHLAs were confirmed to contain somatic mutation-derived peptides (neoantigens) that possess excellent tumor specificity. The emergence of neoantigens-targeted TCRs (e.g., KRAS^G12D^, EGFR^T790M/C797S^, H3.3^K27M^, p53^R175H^) [32,33,34,35] allows for the designing TCR-based drugs for the precision targeting of cancers not amenable to current immunotherapies. TCR/aCD3 was one of the TCR-based drugs that has been proven to be effective in redirecting T cells towards pHLAs to kill tumor cells [15,16]. In this study, the chemo-enzymatic conjugation approach was employed to generate TCR/aCD3. As aCD3 is an invariable component of TCR/aCD3, it has the flexibility to generate diverse TCR/aCD3 by altering TCR. Therefore, we can not only efficiently generate tumor-specific TCR/aCD3 to mimic the function of tumor-specific T cell, but we can also avoid the complex ex vivo amplification of T cells in CAR-T or TCR-T therapies.

We used NY-TCR/aCD3 as a representative to evaluate the biological activity of TCR/aCD3. We found that the NY-TCRmut/aCD3 can redirect naïve T cells or ATC to lysis NY/A02 positive tumor cells, even when they have extremely low antigen expression (no more than 50 copies per cell) [36] The antigen-negative cells were ignored under otherwise identical conditions, which demonstrated the superiority of TCRs in specificity. However, NY-TCRwt/aCD3 (~4 μg/mL EC_50_) was ~10 times less effective than NY-TCRmut/aCD3 (~0.4 μg/mL EC_50_), which indicated that the affinity of natural TCRs needs to be improved to optimize the activity of TCR/aCD3. Since the affinity of antibodies is generally much better than that of TCR, researchers have tried to use TCR-mimic antibodies (TCRm) to target pHLA, and dozens of TCRm have emerged in the past few decades [37,38]. Even so, the TCRm-based biologics were rare, which may be caused by the insufficient specificity of TCRm [38]. In addition, some researchers have endeavored to develop techniques for TCR directed evolution [21,39,40], and the most successful phage display-based platform was established by Li Y et al. [21]. Interestingly, Holland CJ et al. proved that affinity-enhanced TCRs maintain the high specificity of natural TCRs, as they are both less cross-reactive and less promiscuous compared to TCRm [41].

Unfortunately, the potency of the NY-TCRmut/aCD3 (~2 nM EC_50_) we generated was inferior to the previous reported (ImmTAC, ~0.1 nM EC_50_ or lower) [15]. T cell activation is primary for activity and can be initiated upon the binding of the TCR to pHLA (“signal 1”) [42]. TCR/aCD3 can trigger T cell activation by mimicking conventional signal 1 [14], but it remains to be optimized, as we found that the activation of T cells by NY-TCRmut/aCD3 was mild and unabiding. Although the important features influencing T cell activation by TCR/aCD3 remain largely unknown, according to previous studies [14], we suspect that the different interdomain configurations and molecular size between NY-TCR/aCD3 and ImmTAC might be the main contributions to the variant potencies. Therefore, the TCR/aCD3 here provided a discriminative format for the study of the mechanism of TCR/aCD3 activation that was very important for rational TCR/aCD3 design. In addition, costimulatory signals (“signal 2”) can further enhance the T cell activation and proliferation that are essential for natural T cell immune response [43]. Recently, Skokos D, et al. [44] and Wu L, et al. [45] introduced signal 2 by activating CD28 to successfully optimize the activity of T-BsAb. Similarly, the activity of NY-TCRmut/aCD3 could be significantly improved without sacrificing specificity when we used the CD3/CD28 pre-activated T cells to replace naïve T cells. Hence, we speculated that combining the costimulatory signal with TCR/aCD3 may provide precision and effective tumor immunotherapy strategies.

Staphylococci injection is widely used in cancer therapy in China, and the main effective component is claimed to be SEC2 [46]. Furthermore, we employed the powerful T cell activation property of SEC2 [22] to improve the potency of TCR/aCD3. To our surprise, we found that the secretions of SEC2-activated T cells can promote HLA-I expression and thus increase the target level, which may further contribute to the improvement of NY-TCR/aCD3 activity. Based on previous studies, we hypothesized that SEC2-activated T cell-secreted cytokines such as IFN-γ may be the mediators for HLA-I upregulation [47,48]. Moreover, SEC2 may optimize the efficacy of CD3-bispecifics in solid tumors, as Scheidt BV, et al. demonstrated that SEB can support CAR-T cells against solid tumors [49]. Combining the class of T cell activating biologics with the emerging class of TCR/aCD3 may provide another strategy for effective tumor immunotherapy.

## 5. Conclusions

We used heterodimeric Fc to assist in the correct assembly of TCRs to achieve the soluble expression of TCRs in mammalian cells, which allows for yielding novel TCR/aCD3 conveniently. Although the potency of the NY-TCR/aCD3 here was inferior to that of our previous report, it provided a discriminative format for the study of the TCR/aCD3 activation mechanism, which was very important for rational TCR/aCD3 design. Moreover, we proposed a strategy of superantigen (e.g., SEC2) combination to successfully make up for the deficiency of TCR/aCD3 activity. Generally, more than 90% of cellular proteins have peptides that can be potentially presented by HLA and can be the targets of TCRs, which contain somatic mutation-derived peptides (neoantigens) [50] and possess excellent tumor specificity. This study provides a feasible strategy to generate neoantigen-target TCR/aCD3 by altering TCR for the precision targeting of cancers not amenable to current immunotherapies.

## Figures and Tables

**Figure 1 biomedicines-09-00790-f001:**
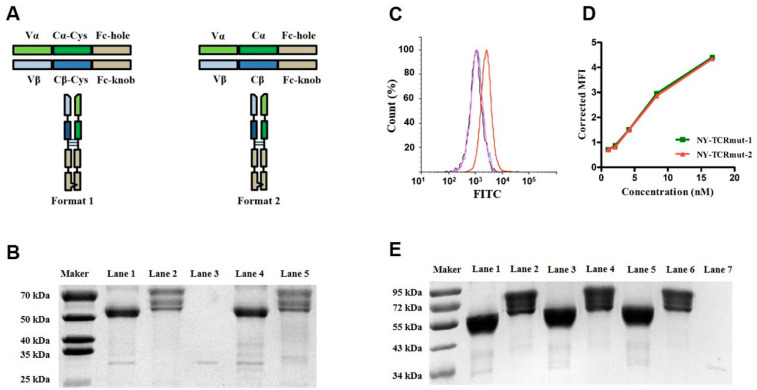
Soluble expressions of TCRs in mammalian cells. (**A**) Two Fc-fused formats of soluble TCR. Format 1: ECDα (with the C terminated cysteine)-*GS*-hinge region-CH2-CH3 (hole)-*G_4_S*-LPETGG and ECDβ (with the C terminated cysteine)-*GS*-hinge region-CH2-CH3 (knob)-*G_4_S*-LPETGG form heterogeneous dimer. Format 2: ECDα (without the C terminated cysteine)-*GS*-hinge region-CH2-CH3 (hole)-*G_4_S*-LPETGG and ECDβ (without the C terminated cysteine)-*GS*-hinge region-CH2-CH3 (knob)-*G_4_S*-LPETGG form heterogeneous dimer. (**B**) Analysis of the NY-TCRmut using SDS-PAGE in denatured condition after being de-glycosylated using PNGase F. Lane 1: deglycosylated NY-TCRmut of format 1. Lane 2: glycosylated NY-TCRmut of format 1. Lane 3: PNGase F (~30 kDa). Lane 4: deglycosylated NY-TCRmut of format 2. Lane 5: glycosylated NY-TCRmut of format 2. α and β chain of deglycosylated NY-TCRmut have a similar molecular size (~50 kDa). (**C**) Analysis of the antigen binding ability and specificity of NY-TCRmut on T2 cells. Black: T2 cells were directly incubated with fluorescent antibody (blank). Blue: T2 cells incubated with irrelevant peptide and NY-TCRmut of format 1 (control). Purple: T2 cells incubated with irrelevant peptide and NY-TCRmut of format 2 (control). Red: T2 cells incubated with relevant peptide and NY-TCRmut of format 1. Green: T2 cells incubated with relevant peptide and NY-TCRmut of format 2. (**D**) The binding ability of the two NY-TCRmut with A375-NY cells. (**E**) Analysis of the TCRs from format 1 using SDS-PAGE in denatured condition after being de-glycosylated by PNGase F. Lane 1: deglycosylated p53^R175H^-TCR. Lane 2: glycosylated p53^R175H^-TCR. Lane 3: deglycosylated Tax-TCR. Lane 4: glycosylated Tax-TCR. Lane 5: deglycosylated NY-TCRwt. Lane 6: glycosylated NY-TCRwt. Lane 7: PNGase F.

**Figure 2 biomedicines-09-00790-f002:**
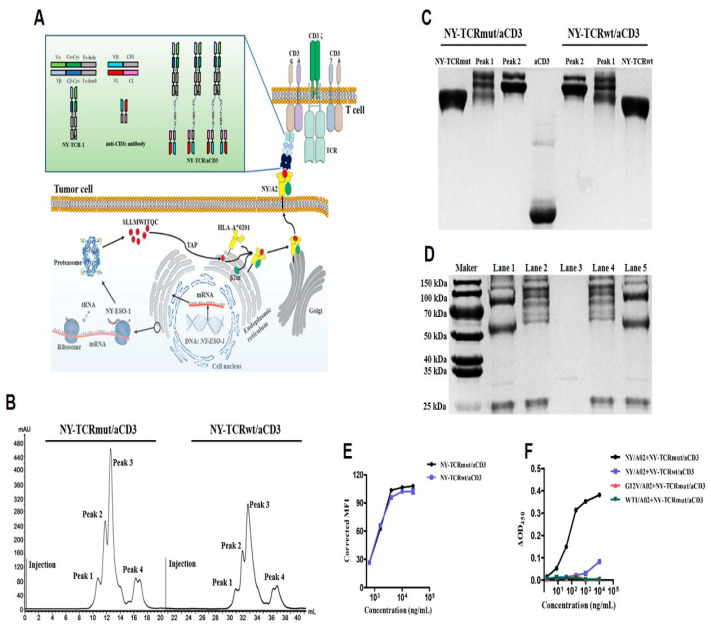
Generation and analysis of NY-TCR/aCD3. (**A**) NY/A02 as the target of TCR/aCD3. Peptides (SLLMWITQC) derived from proteasome digestion are presented to the cell surface by HLA-A*0201, where they are recognized by NY-TCR as the pHLA(NY/A02). NY-TCR/aCD3 was generated through chemo-enzymatic conjugation that can mediate T cell immune response by simultaneously recognizing T cells and tumor cells. (**B**) Purification of NY-TCR/aCD3 using molecular exclusion chromatography. Four peaks were observed, and the products of Peak 1and Peak 2 were collected for further analysis. (**C**) Analysis of the products of Peak 1and Peak 2 using SDS-PAGE in non-denatured condition. The composition of Peak 1is mixed. The molecular size of the two components of Peak 2 is larger than that of NY-TCR, suggesting that one NY-TCR may successfully conjugate with one or two aCD3. (**D**) Analysis of the products of Peak 2 by SDS-PAGE in denatured condition after being deglycosylated using PNGase F. Lane 1: deglycosylated NY-TCRmut/aCD3. Lane 2: glycosylated NY-TCRmut/aCD3. Lane 3: PNGase F. Lane 4: glycosylated NY-TCRwt/aCD3. Lane 5: deglycosylated NY-TCRwt/aCD3. Glycosylated NY-TCR/aCD3 have a variety of components with different molecular sizes, and deglycosylated NY-TCR/aCD3 contain three components: α or β chain of NY-TCR (~50 kDa), aCD3-H conjugated α or β chain of NY-TCR (~75 kDa), and aCD3-L (~23 kDa). (**E**) Binding ability analysis of NY-TCR/aCD3 to Jurkat cells (CD3ε^+^, n = 3). NY-TCRmut/aCD3 and NY-TCRwt/aCD3 have a similar affinity to CD3ε. (**F**) Affinity analysis of NY-TCR/aCD3 to NY/A02 and irrelevant pHLA (n = 3). NY-TCRwt/aCD3 had a weak affinity to NY/A02, while the affinity between NY/A02 and NY-TCRmut/aCD3 was significantly higher. NY-TCRmut/aCD3 did not show any affinity to irrelevant pHLA under otherwise identical conditions.

**Figure 3 biomedicines-09-00790-f003:**
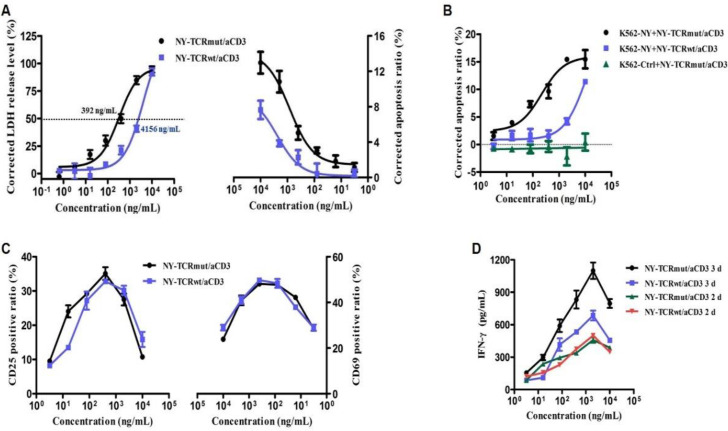
NY-TCR/aCD3 activated and redirected naïve T cells to specifically kill NY/A02^+^ tumor cells. (**A**) Lysis of A375 or A375-eGFP cells over 48 h using PBMC in the presence of titrated concentrations of NY-TCR/aCD3. The lysis was determined by cell apoptosis assay (n = 3, right) or LDH release assay (n = 3, left). (**B**) Lysis of K562-based tumor cells over 48 h using PBMC in the presence of titrated concentrations of NY-TCR/aCD3 (n = 3). The lysis was determined by cell apoptosis assay. (**C**) T cell activations were marked as CD69^+^ or CD25^+^ after treatment with NY-TCR/aCD3 for 72 h (n = 3). CD69^+^ and CD25^+^ T cells significantly increased with different concentrations of NY-TCR/aCD3, but the increased amplitude decreased with high concentrations of NY-TCR/aCD3. (**D**) T cell activations were showed using IFN-γ assay (n = 3). IFN-γ secretion significantly increased in a time-dependent manner with different concentrations of NY-TCR/aCD3, but it decreased at 10 μg/mL, and NY-TCRmut/aCD3 was able to induce more IFN-γ secretion than NY-TCRwt/aCD3 in 72 h. Data are means ± SEM.

**Figure 4 biomedicines-09-00790-f004:**
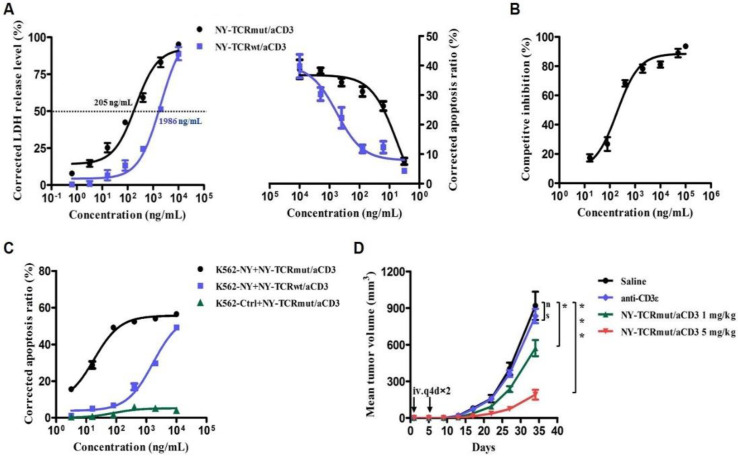
NY-TCR/aCD3 redirected ATC to specifically kill NY/A02^+^ tumor cells. (**A**) Lysis of A375 or A375-eGFP cells over 24 h caused by ATC in the presence of titrated concentrations of NY-TCR/aCD3. Lysis was determined through cell apoptosis assay (n = 3, right) or LDH release assay (n = 3, left). (**B**) Competitive inhibition of NY-TCRmut/aCD3 activity by NY/A02 (n = 3). The lysis level was determined by LDH release assay and competition inhibition = LDH concentration of the test group/LDH concentration of the control group (0 ng/mL NY/A02). (**C**) Lysis of K562-based tumor cells 24 h caused by ATC in the presence of titrated concentrations of NY-TCR/aCD3 (n = 3). Lysis was determined through cell apoptosis assay. (**D**) In vivo efficacy of NY-TCRmut/aCD3 in the beige-SCID xenograft model (n = 5). Beige-SCID mice engrafted with A375 cells (2 × 10^6^) and ATC (4 × 10^6^) were treated with NY-TCRmut/aCD3 with a dose of 1 mg/kg and 5 mg/kg. The blank group and control group mice were dosed with saline and anti-CD3ε antibody, respectively. At day 34, the difference between the 5 mg/kg group compared to the blank group was highly significant (*p* = 0.0003). The difference between the 1 mg/kg group compared to the blank group was significant (*p* = 0.0313), and there was no significant difference between the control group and blank group (*p* = 0.5343). Data are means ± SEM. *p* < 0.05, * *p* < 0.01, *** *p* < 0.001, n.s *p* > 0.05.

**Figure 5 biomedicines-09-00790-f005:**
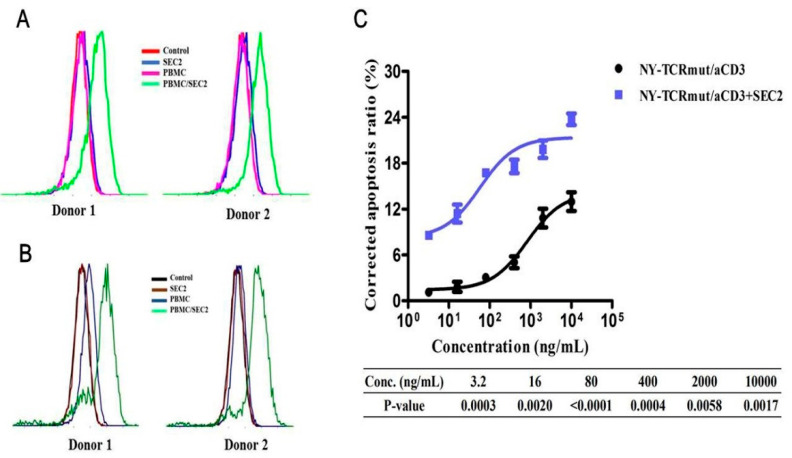
SEC2 enhanced the anti-tumor activity of NY-TCRmut/aCD3. (**A**) The HLA-A*02 expression level in A375 cells was up-regulated after being stimulated by the secretory substances of SEC2-activated T cells. The HLA-A*02 expression level was determined using flow cytometry, and the shift of the peaks to the right indicated the higher expression level of HLA-A*02. The PBMC were from two donors. (**B**) The NY/A02 expression level on the A375-NY cell surface was up-regulated after being stimulated by the secretory substances of SEC2-activated T cells. The NY/A02 expression level was determined using flow cytometry, and the shift of the peaks to the right indicated the higher expression level of NY/A02. The PBMC were from two donors. (**C**) Lysis of A375-eGFP cells over 48 h caused by PBMC in the presence of titrated concentrations of NY-TCRmut/aCD3 with or without 2 ng/mL SEC2 (n = 3). Lysis was determined using cell apoptosis. NY-TCRmut/aCD3 was able to redirect the lytic activity of T cells more effectively at different concentrations with 2 ng/mL SEC2.

## Data Availability

The data presented in this study are available upon request from the corresponding author.

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
