# Peer review of "Soluble Expression of Fc-Fused T Cell Receptors Allows Yielding Novel Bispecific T Cell Engagers"

_biomedicines, 2021, doi:10.3390/biomedicines9070790_

Round 1
Reviewer 1 Report
The authors of the present manuscript investigated a highly important area of cancer therapy. In the introduction the authors described the currently available and clinically applied immuno-therapeutic thechniques that have a great variety of side effects and the treatments are not always effective. The authors set out to develop a technique to increase cytotoxic T cell based approach for an immunotherapy-like anti-cancer treatment. They have used a great variety of techniques to achieve their goal. The initial technology was described in an article in Nat. Biotechnol. 2005, 23, 349–354 that the authors improved and modified in their study.
Critical comments
Due to the large number of techniques it would be essential for the readers to see an overall schematic presentation of the protocols and a shcematic figure of the summary of the research results. Especially so, as the language of the manuscript is hard to follow. In many cases the sentences are either too long or simply don't make any sense. The reader is often left to guess what the authors meant.
Overall the manuscript is lacking clarity. The authors go into great detail about several aspects of the molecular design of the soluble TCR and completely neglect some much less obvious other aspects.
Just an example: While the authors explain in the results why we potentially see three bands in the gel instead of two, they leave the readers guessing how and why they selected the A374 cancer cell line for their experiments?
Although it is clear why the authors used mammalian cells for the production of the soluble TCR/aCD3 (and went into a far too detailed explanation), it remained an enigma for the reviewer how would this process become useful in clinical application? The authors should explain how they think the protein fragment analysis would take place to target neoantigens by TCRs in the presence of various mutations (e.g. KRASG12D, EGFRT790M/C797S, 34H3.3K27M, p53R175H) of a great variety of tumors in a clinical setting. Especially so, as tumors are frequently change during tumorigenesis and aquire additional mutations that would require further selection and adaptation of the initially selected TCR/aCD3.
Admittedly, the strategy of using TCR/aCD3 all by itself was not highly effective and needed the stimulatory influence of a superantigen. It would be important to show what kind of cytokines were produced in the presence of SEC2 that stimulated the otherwise ineffective TCR/aCD3.
Also discussion about the necessity of a superantigen in a clinical application would also be useful. Especially so, as superantigens are potent and aspecific activators of the immune system.
Author Response
Critical comments
- Due to the large number of techniques it would be essential for the readers to see an overall schematic presentation of the protocols and a schematic figure of the summary of the research results. Especially so, as the language of the manuscript is hard to follow. In many cases the sentences are either too long or simply don't make any sense. The reader is often left to guess what the authors meant.
Response:
Thanks for the reviewer’s suggestion. We have modified the manuscript to clear describe the protocol and research results as suggested by the reviewer. Firstly, we have added easy-to-understand graphical abstract to give a brief overview of the research (attached at the end of the revised version). Secondly, we also modified the language of the manuscript, and an English native speaker has helped us to improve the phrasing. The changes have been marked in red in the revised version.
- Overall the manuscript is lacking clarity. The authors go into great detail about several aspects of the molecular design of the soluble TCR and completely neglect some much less obvious other aspects.
Just an example: While the authors explain in the results why we potentially see three bands in the gel instead of two, they leave the readers guessing how and why they selected the A375 cancer cell line for their experiments?
Response:
Thank you for the reviewer’s suggestion. The molecular design of the soluble TCR is of critical important for the generation of TCR/aCD3 and is the basis of this research. We apologize for not explaining the other aspects clearly, and we have modified the expression in the revised manuscript. In addition, we selected the A375 cell line for our experiments because it is naturally antigen-positive cell line. Although we have described all the cell lines involved in this research in the supplementary material, we neglected the reason for cell line selection in the manuscript, which made it difficult for reader to understanding the research results. We have added brief information about the cell lines to the manuscript (section 3.3) as suggested by the reviewer.
- Although it is clear why the authors used mammalian cells for the production of the soluble TCR/aCD3 (and went into a far too detailed explanation), it remained an enigma for the reviewer how would this process become useful in clinical application?
Response:
Thanks for the reviewer’s suggestion. In the revised version, we simplify the explanation for the selection of mammalian cells to produce TCR/aCD3. The advantage of clinical application is one of the most important directions of drug development. A lot of factors can affect the clinical application of drugs, such as drug yield, easy to operate, drug stability, pharmacokinetics, and so on. In the manuscript, we analyzed the advantages of the mammalian cell-produced TCR/aCD3 compared with bacteria-produced TCR/aCD3 in terms of drug yield, operation and safety, which indicated its potential advantages in clinical application. Of course, other clinical advantages not researched here also deserve attention in future studies.
- The authors should explain how they think the protein fragment analysis would take place to target neoantigens by TCRs in the presence of various mutations (e.g. KRASG12D, EGFRT790M/C797S, H3.3K27M, p53R175H) of a great variety of tumors in a clinical setting. Especially so, as tumors are frequently change during tumorigenesis and acquire additional mutations that would require further selection and adaptation of the initially selected TCR/aCD3.
Response:
Thanks for the reviewer’s critical comment. Currently, many studies have focused on the identification of neoantigens and related methods have been reported. However, there are still difficulties in moving from mutation discovery to identification of neoantigens. But, the KRASG12D, EGFRT790M/C797S, H3.3K27M, and p53R175H we mentioned in the manuscript have been identified as target of TCR in several studies (We have quoted relevant references in the manuscript). In addition, the drug resistance proposed by the reviewer does exist (N. Engl. J. Med. 2016, 375:2255-2262), and it is urgently to be solved. But, targeting neoantigens via TCR has also been shown to have clinical benefit in some patients (N. Engl. J. Med. 2016, 375:2255-2262, Science 2016, 352:1337-1341, Cancer Treat. Rev. 2019, 77: 35-43). Furthermore, some researchers have suggested that targeting key mutations associated with tumor development (such as KRASG12D, EGFRT790M/C797S, H3.3K27M, and p53R175H) may reduce the risk of drug resistance (Nat. Rev. Cancer 2020, 20:555-572).
- Admittedly, the strategy of using TCR/aCD3 all by itself was not highly effective and needed the stimulatory influence of a superantigen. It would be important to show what kind of cytokines were produced in the presence of SEC2 that stimulated the otherwise ineffective TCR/aCD3.
Response:
Thanks for the reviewer’s critical comment. SEC2 can stimulate numerous T-cells for proliferation and to release massive amounts of cytokines like IL-2, IL-4, TNF and IFN-γ (Int. J. Cancer 1993, 54:482-488). Several cytokines like TNF-α (Neurochem. Res. 2013, 38:2295-2304) and IFN-γ (Int. Rev. Immunol. 2009, 28:239-260) can upregulate the expression level of HLA-I. Therefore, we speculated that a variety of cytokines were involved in the optimization of TCR/aCD3 activity. In this research, we also proved that SEC2 stimulates T cells to release IFN-γ (Fig S4A), suggesting that IFN-γ may be one of the cytokines involved in the optimization of TCR/aCD3 activity.
- Also discussion about the necessity of a superantigen in a clinical application would also be useful. Especially so, as superantigens are potent and aspecific activators of the immune system.
Response:
Thank you for the reviewer’s suggestion. In the revised version, we have added the clinical application of SEC2 in the discussion section (line 47, page 12).

Reviewer 2 Report
Zhao et al in this manuscript has described a new strategy for expression of stable and soluble TCR in mammalian cells that can yield novel bispecific T cell engagers. It is an important study in the scientific field and can be exploited for use in precision targeting in cancer. I have following comments.
-Title needs to short and should be reframed.
-in methods section, authors has reported pMH3 plasmid was in our laboratory. Authors should mention the exact source of this plasmid?
-Authors should describe the methods in detail For e.g., in cell binding assays, it is not clear for how long NY TCR/aCD3 and cells were incubated before being analyzed by flow cytometry. Do authors have flow plots, if yes should be included in supplementary?
-Do authors know the percentage of T cells like CD4, CD8 in biological replicates, is it same before using for the experiment? Instead of PBMC which contains all immune cells, I would suggest authors should isolate T cells, expand them and use them for activation or for effector cell function or any other assay.
-It is not clear how much time after engraftment of A375 cells in SCID mice was ATC injected? What was the tumor size?
-Why do authors have measured CD25+ as a marker of activation? CD25+ is a marker of Tregs and play a role in immunosuppression.
-Instead of writing days for in vitro incubation times, authors should change it to hours which is standard way of writing it.
-
Author Response
- Title needs to short and should be reframed.
Response:
Thanks for reviewer’s suggestion, and the title has been shortened and reframed. TCR/aCD3 has been shown to have clinical efficacy in immunologically cold solid tumors, and provide potent and highly specific access to the vast landscape of intracellular targets (Cancer Treat. Rev. 2019, 77: 35-43). However, preparation of accredited TCR/aCD3 remains challenging. In this research, we aimed to explore feasible strategies for the preparation of TCR/aCD3, and the activity of TCR/aCD3 was evaluated to prove the feasibility of the strategy. Therefore, we think that the title of this article should focus on TCR/aCD3 preparation. In addition, aCD3 is the main choice of T cell engagers at present, so it does not hinder reader’s understanding when we removed “through pairing with the anti-CD3 antibody”.
- In methods section, authors have reported pMH3 plasmid was in our laboratory. Authors should mention the exact source of this plasmid?
Response:
We have added the exact source of pMH3 plasmid in the methods section (line 9, page 3) of the revised manuscript.
- Authors should describe the methods in detail For e.g., in cell binding assays, it is not clear for how long NY TCR/aCD3 and cells were incubated before being analyzed by flow cytometry. Do authors have flow plots, if yes should be included in supplementary?
Response:
Thanks for the reviewer’s suggestion. We have detailed the description of the methods in the hope of making it more clear to the reader, and the changes have been marked in red in the revised version.
- Do authors know the percentage of T cells like CD4, CD8 in biological replicates, is it same before using for the experiment? Instead of PBMC which contains all immune cells, I would suggest authors should isolate T cells, expand them and use them for activation or for effector cell function or any other assay.
Response:
Thanks for the reviewer’s suggestion. The PBMC from one donor was stored separately after isolated in the same batch. Although the PBMC from two donors were used in this research (the percentage of CD4+ or CD8+ T cells among different donors was different), the same donor’s PBMC was used to evaluate the activity of TCR/aCD3, so the percentage of T cells like CD4, CD8 is same before using for the experiment.
Isolation of T cells for assay is a meaningful suggestion, which makes it easier to control variables and understanding the T cell response. However, there are two main considerations that led us to use PBMC in this research. Firstly, it is expensive to isolate T cells from PBMC, and T cell activity may be affected during isolated operation. In addition, we preliminary evaluated the activity of TCR/aCD3 to prove the feasibility of the strategy for TCR/aCD3 preparation, and we believe that PBMC can meet the needs of the assays. Secondly, although other immune cells may affect the function of T cells, it is more realistic to represent the immune response in vivo. In the subsequent pre-clinical evaluation of TCR/aCD3 activity, we will seriously consider the suggestions of the reviewer and use PBMC or isolated T cells more reasonably for assays.
- It is not clear how much time after engraftment of A375 cells in SCID mice was ATC injected? What was the tumor size?
Response:
Thanks for the reviewer’s suggestion. It's our fault that we did not explain clearly the construction of the xenograft tumor model in the manuscript, and we have revised it in the method section (line 18, page 5). A375 cells and ATC were mixed, and then engrafted subcutaneously in the mice. Mice were treated with different drugs 1 h after engraftment (No tumor was seen at this time). The tumor volume was calculated according to the formula (tumor volume= length×width2/2), which was mentioned in the method section (section 2.10).
- Why do authors have measured CD25+ as a marker of activation? CD25+ is a marker of Tregs and play a role in immunosuppression.
Response:
Thanks for the reviewer’s suggestion. CD25 also called IL-2Rα, which is one of the subunits of IL2 receptor. Indeed, CD25 is one of the markers of Tregs. However, it also plays an important role in T cell activation and proliferation, and the increased expression of CD25 has been used as intermediate or late marker of T cell activation for many researches (J. Immunol. Methods 2004, 293:127-142, Annu. Rev. Immunol. 2008, 26:453-479, Sci. Transl. Med. 2015, 7:287ra70). There are several markers of T cell activation, and we chose CD69 (early marker), CD25 and IFN-γ as the markers for a more comprehensive evaluation of T cell activation.
- Instead of writing days for in vitro incubation times, authors should change it to hours which is standard way of writing it.
Response:
Thank you for the suggestion, and we have changed the days to hours for in vitro incubation time (the changes have been marked in red in the revised version).

Round 2
Reviewer 1 Report
Thank you for the presented changes that improved the manuscript. The native speaker who helped with the revised version ought to read the abstract as well.
Also, I cannot locate the graphical abstract.
Reviewer 2 Report
Overall, authors have significantly improved the manuscript but have missed supplemenatry flow cytometry plots for NY-TCRs or NY-TCR/aCD3 binding affinity on cells.